# Strong Biofilm Formation and Low Cloxacillin Susceptibility in Biofilm-Growing CC398 *Staphylococcus aureus* Responsible for Bacteremia in French Intensive Care Units, 2021

**DOI:** 10.3390/microorganisms10091857

**Published:** 2022-09-16

**Authors:** Nathalie van der Mee-Marquet, Sandra Dos Santos, Seydina M. Diene, Isabelle Duflot, Laurent Mereghetti, Anne-Sophie Valentin, Patrice François

**Affiliations:** 1Centre d’Appui pour la Prévention des Infections Associées aux Soins (CPias) de la Région Centre Val de Loire, Hôpital Bretonneau, Centre Hospitalier Régional Universitaire, 37044 Tours, France; 2Faculté de Pharmacie, Microbes Evolution Phylogeny and Infections, IHU-Méditerranée Infection, Aix-Marseille Université, 13005 Marseille, France; 3Service de Bactériologie, Virologie et Hygiène, Hôpital Trousseau, Centre Hospitalier Régional Universitaire, 37044 Tours, France; 4Department of Medicine, University of Geneva Hospitals, 1205 Geneva, Switzerland

**Keywords:** *Staphylococcus aureus*, intensive care, CC398, bacteremia, biofilm, cloxacillin, antibiotic tolerance

## Abstract

A prospective 3-month study carried out in 267 ICUs revealed an *S. aureus* nosocomial bacteremia in one admitted patient out of 110 in adult and pediatric sectors, and in one out of 230 newborns; 242 *S. aureus* bacteremias occurred during the study, including 7.9% MRSA-bacteremias. In one ICU out of ten, the molecular characteristics, antimicrobial susceptibility profiles and biofilm production of the strains responsible for *S. aureus* bacteremia were studied. Of the 53 strains studied, 9.4% were MRSA and 52.8% were resistant to erythromycin. MLST showed the predominance of CC398 (37.7% of the strains) followed by CC8 (17.0%), CC45 (13.2%) and CC30 (9.4%). The *lukF/S* genes were absent from our isolates and *tst-1* was found in 9.4% of the strains. Under static conditions and without exposure to glucose, biofilm production was rare (9.4% of the strains, without any CC398). The percentage increased to 62.3% for strains grown in broth supplemented with 1% glucose (including 7 out of 9 CC8 and 17 out of the 20 CC398). Further study of the CC398, including whole genome sequencing, revealed (1) highly frequent patient death within seven days after CC398 bacteremia diagnosis (47.4%), (2) 95.0% of the strains producing biofilm when exposed to sub-inhibitory concentrations of cloxacillin, (3) a stronger biofilm production following exposure to cloxacillin than that observed in broth supplemented with glucose only (*p* < 0.001), (4) a high minimum biofilm eradication concentration of cloxacillin (128 mg/L) indicating a low cloxacillin susceptibility of biofilm-growing CC398, (5) 95.0% of the strains carrying a ϕSa-3 like prophage and its particular evasion cluster (i.e., yielding *chp* and *scin* genes), and (6) 30.0% of the strains carrying a ϕMR11-like prophage and yielding a higher ability to produce biofilm. Our results provide evidence that active surveillance is required to avoid spreading of this virulent staphylococcal clone.

## 1. Introduction

*S. aureus* possesses numerous virulence factors facilitating tissue colonization, immune evasion and tissue destruction. One of its defense mechanisms is the capacity to form biofilms. Bacteria embedded in biofilms are resistant to host immune response and difficult to eradicate with antibiotic [1,2]. Biofilm-forming capacity is a virulence determinant in the development of catheter-related infections, and the effective treatment of staphylococcal infections that share such features has become a challenge [3]. 

Described initially in livestock, *S. aureus* of clonal complex 398 (CC398) are increasingly responsible for bacteremia in humans living in animal-free environments [4,5,6,7]. Whole genome sequencing analysis of CC398 strains have demonstrated that livestock-associated and emerging strains differ by their prophage content [8,9]. The emerging strains usually carry ϕSa-3 and ϕMR11-like prophages, which have been shown to be involved in immune escape, adhesion to host cells and extracellular matrix components, along with epithelial cell invasion, i.e., factors contributing to an increased ability of the bacteria to colonize and infect the host [10]. 

Enhanced biofilm formation ability was recently described in emerging CC398 strains in China [11]. Facing the epidemiological changes observed with this clone in human clinical settings, epidemiological data remain scarce. We conducted a prospective incidence-study of nosocomial bacteremia in 267 intensive care units (ICUs), and analyzed the molecular characteristics and antimicrobial susceptibility profiles of the *S. aureus* strains responsible for bacteremia in one ICU out of ten. We sequenced the genome of the CC398 strains, analyzed their prophage content, studied their ability to produce biofilm following exposure to sub-inhibitory concentrations of cloxacillin, the first line treatment for patients suffering from MSSA bacteremia, and determined their cloxacillin susceptibility in biofilm, where applicable.

## 2. Materials and Methods

### 2.1. Nosocomial Bacteremia Survey, Data Collection and Analysis

In each participating ICU, a 3-month surveillance of nosocomial bacteremia was carried out between 1 January and 15 August 2021. A protocol derived from that of the ECDC HAI light surveillance protocol was used (https://www.ecdc.europa.eu/sites/default/files/documents/HAI-Net-ICU-protocol-v2.2_0.pdf; accessed on 1 January 2021). During the survey period, each positive blood culture was analyzed to determine whether it was associated with a nosocomial bacteremia. The origin of the bacteremia was determined using clinical and biological data. Data were collected (HCF data, patient characteristics (sex, age, birth weight for newborns, the severity index, the immune status, the type of cancer where applicable, and death within 7 days of bacteremia diagnosis), characteristics of the bacteremia (origin (skin (primary cutaneous form or superinfection of a skin wound), surgical site, lungs, urinary tract, intravascular device, intra-abdominal or digestive tract) and microorganisms). Data processing, validation of the database and data analysis were carried out by the national team using R software (version 3.6.1 on Ubuntu; General Public Licence; Vienna, Austria). For the variables studied, percentages were calculated from the numbers, without taking missing data into account. Incidence rates were generated according to patient-days and admitted patients. Here, we report the data regarding *S. aureus* bacteremias. 

### 2.2. Microbiological Study

Local infection control teams were asked to send the *S. aureus* strains isolated from bacteremia to the central lab using transport swabs (Copan Italia SPA, Brescia, Italy). The strains were tested for antimicrobial drug susceptibility using the AST-P631 vitek card (oxacillin, cefoxitin, gentamicin, tobramycin, ofloxacin, erythromycin, linezolid, teicoplanin, vancomycin, tetracycline, nitrofurantoin, fusidic acid, rifampicin, trimethoprim-sulfamethoxazol, fosfomycin; bioMérieux, France), Etest^®^ (daptomycin, mupirocine; bioMérieux, Marcy-l’Étoile, France) and Eucast guidelines (http://www.eucast.org/; accessed on 1 January 2021). The presence of *mecA/C*, *mupA/B*, *qacAB/C*, *tst-1*, and *lukF/S* genes was determined using PCR. Strain genomic diversity was studied using MLST (https://pubmlst.org/organisms/staphylococcus-aureus; accessed on 1 January 2021). Purified genomic DNA for CC398 strains was sequenced on the Illumina HiSeq (Illumina, San Diego, CA, USA) using 100 base-pair paired-ends read and bar code strategy according to the Nextera XT kit (Illumina, San Diego, CA, USA), following the manufacturer’s recommendations [12]. The ability to produce biofilm was assessed under static conditions using the method of Christensen [13]. Bacteria were grown at 37 °C in Tryptic Soy broth (TSB) or TSB supplemented with 1% D-(+)-glucose (Sigma Aldrich), or TSB supplemented with 1% D-(+)-glucose and cloxacillin at a concentration of MIC/4. After 48 h of growth, the plates were washed three times with Phosphate-buffered saline, prior to staining with a 0.4% crystal violet solution. Each strain was tested three times. A biofilm-positive phenotype was defined as an optical density at 595 nm twice that of the negative control, and the strains were divided into four categories: no biofilm producer, weak, moderate and strong producers [14]. To assess a decrease in susceptibility to cloxacillin, bacteria in biofilm were tested using the Calgary biofilm device (CBD, Innovatech Inc., Edmonton, Alberta, Canada) to determine the minimum biofilm eradication concentration of cloxacillin (MBEC) following the manufacturer’s recommendations [15]. The biofilm inhibitory concentrations were defined as the lowest concentrations of drug resulting in an OD650 nm difference of ≤10% of the mean of two positive control well readings. 

### 2.3. Statistical Analysis and Ethics Approval

For categorical variables, we used Pearson’s chi-squared test to compare groups. All analyses were two-tailed and a *p* < 0.05 was considered significant. We used Stata version 10.0 software (Stata Corp., College Station, TX, USA) for statistical analysis. 

Ethics approval for the survey was obtained from the Réseau national de Prévention des Infections Associées aux soins (REPIAS), Santé Publique France national agency. Written informed consent was exempted, since the study focused on bacteria and patient intervention was not required.

## 3. Results

### 3.1. Nosocomial bacteremia 3-Month Survey

The survey was carried out in 267 ICUs at 212 French hospitals. Monitoring focused on 3668 beds (i.e., 52% of French ICU beds) and covered 313,891 patient-days (PDs). A total of 1931 nosocomial bacteremias occurred during the study in the ICUs, and out of these bacteremias, 242 were *S. aureus* bacteremias (12.5%), including 7.9% MRSA-bacteremias (19/240; 2 nk). An *S. aureus* bacteremia was detected in one admitted patient out of 110 in adult and pediatric sectors, and in one out of 230 newborns (Table 1).

The characteristics of the infected patients and their bacteremias are presented in Table 2. The 242 patients suffering *S. aureus* bacteremia had frequent immunodepression (12.3%), cancer (13.6%), and COVID-19 infection (61.5%). In adults, the IGS II severity score was 40.0. The major sources of the *S. aureus* bacteremias were the lungs (50.0%) and intravascular-catheters (62; 25.6%). Death within the first week following infection diagnosis was notified in 69 cases (28.8%).

### 3.2. Microbiological Study

Of the 267 ICUs participating in the survey, 30 took part in the microbiological study (11.2%). Despite this modest ICU participation in our study, the number of *S. aureus* bacteremias detected in these 30 ICUs represented 22.7% of the whole group of 242 bacteremias. In addition, whereas 55 *S. aureus* bacteremias occurred during the study in these 30 ICUs, 53 *S. aureus* strains were available (96.4%). As there were no significant differences regarding the source of the 53 bacteremias compared with those of the 242 from the national level, we considered the 53 strains representative of the national set of *S. aureus* bacteremias (Table 2). 

Of the 53 *S. aureus* strains, 5 were MRSA (9.4%) and 28 were resistant to erythromycin (52.8%) (Table 3). MLST revealed 20 CC398 strains (37.7%), 9 CC8 (17.0%), 7 CC45 (13.2%), 5 CC30 (9.4%), and 12 in 6 other CCs. *mupA*, *qacAB*, and *qacC* were carried by 2 CC8, 1 CC398, and 1 CC8 strain, respectively; *tst-1* was found in 5 strains (9.4%); the *lukF*/*S* genes encoding PVL were absent from our isolates. An association was found (1) between CC8 and methicillin resistance (*p* = 0.039), fusidic acid resistance (*p* < 0.001), and mupirocin resistance (*p* = 0.026), (2) between CC398 strains and erythromycin resistance (*p* < 0.001), and (3) between CC30 strains and the *tst-1* gene (*p* < 0.001). 

Under static conditions and without exposure to glucose, a biofilm production was observed in 5 strains (9.4%), including one strong producer. The percentage increased to 60.4% for strains grown in broth supplemented with 1% glucose, including 6 strong producers (18.7%). No association was found between any clone and a particular bacteremia source. Patient death within seven days after bacteremia diagnosis was most frequent with bacteremia associated with a CC398 or a CC45 strain (47.4% and 42.9%, respectively) compared to bacteremia associated with strains from other CCs (11.5%; NS).

The 20 CC398 strains were sequenced and studied with a previously studied collection of 23 CC398 strains (i.e., six livestock-associated strains and 17 emerging strains; Figure 1) [16]. Among the 20 strains recovered during the present study, 19 carried a ϕSa-3 prophage (95.0%) and its particular evasion cluster (8). The ϕSa-3 prophages were inserted in 18 cases into the virulence gene *hlb*, and in the remaining strain, in *ebh*, a gene encoding for the giant surface anchored protein Ebh that has been associated with *S. aureus* complement resistance [17]. Six strains carried a ϕMR11-like prophage (30.0%), all inserted into the *smpB* virulence gene. Two carried other prophage features (10.0%), and one strain, responsible for a case of endocarditis in a 74 year-old female, did not carry any prophage elements. 

Biofilm production of CC398 strains was further studied in the presence of sub-inhibitory concentrations of cloxacillin. Whereas glucose-induced biofilm production was demonstrated in 85.0% of the CC398 strains, all but one strains produced biofilm in broth supplemented with 1% glucose and cloxacillin at a concentration of MIC/4 (95.0%; NS). Strong producers were more frequent following exposure to sub-inhibitory concentrations of cloxacillin (18/19; 94.7%) than in the case of strains grown with glucose only (3/17; 17.6%; *p* < 0.001), suggesting that cloxacillin at sub-inhibitory concentration may have increased biofilm production in CC398 strains.

Biofilm eradication assay of CC398 strains. The minimum biofilm eradication concentration (MBEC) of cloxacillin, determined for the 20 CC398 strains, ranged from 0.5 to >256 mg/L according to the strain (Figure 1), with a median value of 128. Strains carrying a ϕMR11-like prophage more frequently presented a MBEC > 64 mg/L rather than strains lacking this prophage (1/5 vs. 6/14; NS), suggesting an impact of lysogeny by ϕMR11-like on the susceptibility to cloxacillin of sessile CC398 bacteria.

## 4. Discussion

To our knowledge, this is the first multicenter study depicting the incidence of CC398 *S. aureus* bacteremia in ICUs. The survey revealed one nosocomial bacteremia for 230 newborns, and twice more in adult and pediatric patients, a situation close to the one described in a French ICU in 2011, where a 5-month study revealed 1 case of CC398 nosocomial bacteremia in 89 patients [7]. We observed one third of bacteremias associated with a CC398 strain. This prevalence, obtained in the ICU setting, was higher than that shown in a recent study carried out in all the departments of 17 Spanish hospitals (i.e., medical, surgical, and ICU) [18], in which CC398 strains represented 4.3% of the *S. aureus* recovered from bacteremias during a 6–12-month period in 2018–2019. Due to the similarity of the characteristics of both infected patients and bacteremia in the 30 ICUs and in the 267 ICUs that participated in the French survey, we believe our results may reflect the general situation. Further studies should be carried out, both inside and outside the ICU, to investigate the current incidence of the CC398 clone in hospitals worldwide. 

As usual in Europe [18,19], the CC398 strains responsible for bacteremia were mostly of *spa*-types t1451 and t571, methicillin and tetracycline-susceptible and resistant to erythromycin. The lack of MRSA is reassuring, as is the rarity of strains carrying a *qac* gene, as recent studies mostly carried out in Asia have described an increasing incidence of severe infections caused by livestock-independent MRSA [11,20,21]. 

A high mortality rate was associated with CC398 bacteremias (47.4%), a concordant result with previous studies [22]. However, in contrast to the strains recently described in China [23] and Australia [24], none of the 20 CC398 studied carried the *lukF/S* or *tst-1* genes in their genome. Patient risk factors such as extreme age, immunosuppression (16.7%), and cancer (15.0%), or other unknown compounds present on mobile genetic elements and that should be carefully studied, might play a major role in infection outcome. Note that the number of events is moderate and would need analysis of a larger number of strains.

WGS confirmed typical prophage content of the CC398 emerging strains, with ϕSa-3 in all but one isolate, and ϕMR11 in one third of these [8,9]. Consequently, the strains carried the putative virulence genes *chp* and *scn* in ϕSa-3 encoding the chemotaxis inhibitor protein (CHIPS) and the staphylococcal complement inhibitor (SCIN), respectively, and *seb* encoding a putative superantigen similar to enterotoxin B in ϕMR11. As previously demonstrated, CC398 lysogens carrying these prophage genes may benefit from the production of the three immune-modulating proteins, CHIPS, SCIN, and SEB, when exposed to conditions favoring prophage induction [25,26]. Analysis of the insertion sites of the prophages identified three different loci in the bacterial genomes studied, located in all cases in bacterial virulence genes (i.e., *hlb*, *smpB*, or *ebh*). As shown with a prophage favoring *S. aureus* colonization of diabetic feet [27], lysogeny of the strains with ß-converting prophages may limit *S. aureus* virulence and favor colonization, and thus bacteremia [28]. In the ICU, where risk factors of infections are numerous, whether patient-related or healthcare-related, it would seem likely that in *S. aureus*, the acquisition of such prophages could contribute to a further increase of the ability to colonize human flora and devices and infect humans.

The ability of *S. aureus* to form biofilm is a significant factor that enhances the pathogenicity of this species [1,3,29]. Concordant with previous studies [3], a minority of our strains was capable of biofilm development under standard conditions in TSB, but enhanced biofilm formation was obtained when strains were grown in broth supplemented with 1% glucose [30]. We confirmed a strong biofilm formation associated with CC8 [31], one of the major lineages associated with catheter-related bacteremia [32]. To our knowledge, our study is the first showing that the CC398 lineage also has a high ability to produce biofilm. 

Due to slow diffusion into the biofilm, biofilm bacteria are exposed to sub-inhibitory concentrations of antibiotics [33]. Previous studies have demonstrated that biofilm formation is increased when *S. aureus* are cultured in the presence of sub-inhibitory concentrations of antibiotics [34], and especially oxacillin, i.e., the antibiotic of choice in the course of severe MSSA infections [35]. Our study is the first to show a strong production of biofilm by CC398 strains following exposure to sub-inhibitory concentrations of cloxacillin. Biofilm helps bacteria to escape immune response, contributes to bacterial persistence in the environment [36,37,38,39], and increases phenotypic resistance to antimicrobials [40]. Biofilm-grown microorganisms have an inherent lack of susceptibility to antibiotics, whereas planktonic cultures of the same organism do not [2,33,41,42,43]. We demonstrated extremely high minimal biofilm eradication cloxacillin concentrations of the CC398 MSSA studied. The resulting tolerance of microbial biofilms to in principle adequate antibiotic therapy may lead to problems in their eradication and in the management of infected patients. Our preliminary data, showing cloxacillin-induced strong production of biofilm by CC398 strains, and a tolerance of *S. aureus* to cloxacillin should be explored further.

The determinants carried by the ϕSa-3 and ϕMR-11 prophages have previously been associated with high levels of transmissibility [10], and high potential to persist and disperse in the hospital environment [7]. In this study, the strains recovered from a same ICU were genetically distant, allowing us a priori to exclude intra-ICU transmission. The design of our study did not allow us to investigate the mechanisms of acquisition of the *S. aureus* strains. The genetic diversity observed in our study argues for (but does not prove) dissemination of the CC398 human-adapted subpopulation in human, and nosocomial bacteremia likely caused by multiple strains colonizing the patients before the hospital stay or acquired during hospital stay. Our results suggest the spread of a virulent clone in the hospital settings, confirming the need to investigate the mechanisms involved with the epidemiological success of CC398 in humans.

## Figures and Tables

**Figure 1 microorganisms-10-01857-f001:**
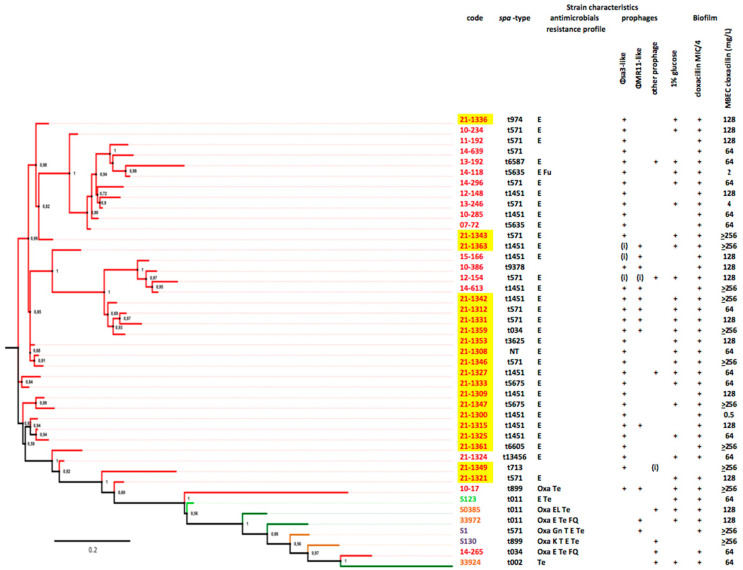
Phylogeny of CC398 strains, prophage content, biofilm production, and MBEC of cloxacillin. Emerging strains (i.e., those from human bacteremia contracted in an animal-free environment) are shown in red, whereas the remaining strains are livestock-associated (animal colonization in light green, animal infection in dark green, and human colonization in yellow). Prophages are indicated by “+” when complete and by “(i)” when incomplete. Note that the emerging strains isolated in 2021 are highlighted in yellow.

**Table 1 microorganisms-10-01857-t001:** *S. aureus* bacteremia incidence rates.

ICU Sector	Bacteremia Incidence (Average Value (Median; Standard Deviation))		
	All Bacteremias	*S. aureus* Bacteremias	
	/1000 patient-days	/1000 patient-days	/100 admitted patients
Adult	5.79 (5.01; 4.63)	1.58 (0.39; 11.01)	0.88 (0.29; 1.53)
Pediatric	3.05 (2.26; 3.38)	0.24 (0.00; 0.55)	0.88 (0.00; 2.88)
Neonatal	2.50 (2.50; 3.78)	0.37 (0.00; 0.76)	0.45 (0.00; 1.02)

**Table 2 microorganisms-10-01857-t002:** Characteristics of the patients suffering from bacteremia.

Patient Characteristics		Patient with Nosocomial Bacteremia					
		All	%	With *S. aureus* Bacteremia			
				All	%	With Studied Strains %	
**N**		1931		242	12.5	53	
**HFC** **category**	univ./region/army	728	37.7	89	36.8	17	31.1
	general	928	48.1	118	48.8	33	62.3
	short stay clinics	262	13.6	31	12.8	3	5.7
	oncology centers	13	0.7	4	1.7		
**Patient category**	adults	1736	89.9	221	91.3	50	94.3
	children	34	1.8	4	1.7	1	1.9
	newborns	161	8.3	17	7.0	2	3.8
**Sex**	males (nk ^1^)	1316 (20)	68.9	151 (1)	62.7	37	74.0
**Age (yr) ^2^**	adults	63.4 (66.0)		63.7 (66.0)		63.8 (64.5)	
	children	2.6 (<1.0)		0.3 (<1.0)		14 (14)	
**Neonatal**	**birth weight (g) ^2^**	1011.2 (840.0)		981.2 (770.0)		1820.0 (1820)	
**Neonatal**	**gestational age (Wk) ^2^**	27.6 (27.0)		27.3 (26.0)		31.5 (31.5)	
**Immunosuppression ^3^ (nk ^1^)**		295 (62)	16.9	30 (9)	13.6	10 (1)	19.2
**Cancer (nk ^1^)**		195 (175)	12.0	26 (19)	12.3	6 (8)	13.3
**Severity score IGS II (adults) ^2^**		45.9 (43.0)		44.4 (40.0)		40.0 (40.0)	
**COVID-19 status (nk ^1^)**		992 (186)	56.8	136 (21)	61.5	32 (7)	69.6
**Patient death ^4^ (nk ^1^)**		533 (27)	28.0	69 (2)	28.8	15 (1)	28.8
**Bacteremia source**							
	catheters	538	27.9	62	25.6	14	26.4
	pneumonia	529	27.4	121	50.0	24	45.3
	urinary tract	115	6.0	1	0.4	1	1.9
	digestive tract	139	7.2	3	1.2		
	other	200	10.3	26	10.7	5	9.4
	not known	410	21.2	29	12.0	9	17.0

^1^ nk not known; ^2^ average value (median value); ^3^ the presence of immunosuppression is defined according to several criteria: 1- either in the presence of true aplasia with <500 circulating polynuclear; 2- either the patient is on immunosuppressive treatment (chemotherapy, radiotherapy, immunosuppressants, long-term and high-dose corticosteroid therapy dose, e.g., for >30 days, recent corticosteroid therapy at a dose >5 mg/kg of Prednisolone for >5 days); 3- either the patient is HIV positive with CD4 < 500/mm^3^, or 4- either the patient has leukemia or lymphoma with PNN < 500/mm^3^. ^4^ Patient death within 7 days after bacteremia diagnosis.

**Table 3 microorganisms-10-01857-t003:** Antimicrobial susceptibility profiles, virulence genes and biofilm production of the 53 *S. aureus* strains, bacteremia source and patient death within 7 days after bacteremia diagnosis, according to sequence type obtained by MLST.

Strain Characteristics		According to MLST										All Strains
		1	5	8	12	15	22	30	45	59	398	
N		2	3	9	1	3	2	5	7	1	20	53
**Antimicrobial susceptibility**												
	**FOX resistance**	1	1	3								5
	**erythromycin resistance**	1		5				2		1	19	28
	**fusidic acid resistance**	1		4								5
	**fluoroquinolone resistance**			2								2
	**fosfomycin resistance**			1								1
	**tetracycline resistance**	1										1
	**kanamycin resistance**	1										1
	** *mupA* **			2								2
	** *qac* **			1							1	2
**Virulence genes**												
	** *tst-1* **							3	2			5
	** *lukF/S* **											0
**Biofilm production**												
	**without glucose 1%**	2	1	1					1			5
	**strong producers**								1			1
	**with glucose 1%**	2	2	7		2			2	1	16	32
	**strong producers**	2		1							3	6
	**with glucose 1% and cloxacillin MIC/4 ^1^**										19	
	**strong producers**										17	
**MBEC cloxacillin ^1,2^ (mg/L)**											128	
**Bacteremia source**												
	**catheters**	1		3	1			3	2	1	3	14
	**pneumonia**		3	4		2	1	1	4		9	24
	**urinary tract**			1								1
	**digestive tract**											
	**endocarditis**										2	2
	**skin and soft tissue**										3	3
	**not known**	1		1		1	1	1	1		3	9
**Death ^3^ (nk ^4^)**				1	2					3		9 (1)

^1^ carried out with the 20 CC398 strains; MIC/4 indicates a concentration of cloxacillin equal to the strains’ MIC value for cloxacillin divided by 4; ^2^ median value; ^3^ patient death within 7 days after bacteremia diagnosis; ^4^ nk not known.

## Data Availability

Not applicable.

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
