# Peer review of "Strong Biofilm Formation and Low Cloxacillin Susceptibility in Biofilm-Growing CC398 Staphylococcus aureus Responsible for Bacteremia in French Intensive Care Units, 2021"

_microorganisms, 2022, doi:10.3390/microorganisms10091857_

Round 1

Reviewer 1 Report

Thank you for the opportunity to review the manuscript “Strong biofilm formation and low cloxacillin susceptibility in biofilm-forming CC398 Staphylococcus aureus responsible for bacteremia in French intensive care units, 2021,” by Nathalie van der Mee-Marquet, et al. The authors performed a study of nosocomial  S. aureus bacteremia (SAB) in 267 French ICUs for 3 months in 2021. There were 1,931 nosocomial bacteria cases recorded. They found that there were 242 cases caused by S. aureus, of which 7.9% were caused by MRSA strains. They chose 10% of the ICUs to perform a more intensive study of SAB and the bacterial isolates causing it. Among the 53 isolates examined, 37.7% were CC398, 17% were CC8, and 13.2% were CC45, with these 3 groups accounting for about 2/3 of all isolates. The percentage of S. aureus belonging to CC398 was remarkably high. Mortality within 7 days of diagnosis among those with CC398 SAB was surprisingly high, at 47.4%. Only 9.4% of the S. aureus isolates were MRSA strains. When glucose was added to growth medium, 62.3% of the isolates formed biofilm, an increase from only 9.4% in the absence of glucose supplementation.  CC398 strains were more likely to form biofilm with cloxacillin broth supplementation compared with glucose supplementation. They also identified a specific prophage more common in strains that were biofilm-formers, the phi-MR11-like prophage.

The authors present data to suggest that biofilm is protective in CC398 MSSA isolates against killing by cloxacillin and that biofilm may be induced by glucose exposure and cloxacillin exposure. The genetic variability among CC398 strains suggests, the authors report, that these strains may have been introduced from community reservoirs before they went on to cause bacteremia. All of the conclusions would be better supported by a larger sample size of S. aureus isolates from SAB.  

The sample is unfortunately very modest, and thus the conclusions can only be considered quite tentative. A larger sample size would enable evaluation for statistical significance of the findings comparing CC398 strains to others and also comparing strains with differing phage content. 

I have several questions and comments for the authors. 

1.     Did the authors test to determine if CC398 was an independent risk factor for death at 7 days?  The mortality rate among CC398 cases was remarkably elevated, but if these cases occurred disproportionately in patients who had other known risk factors for death (older age, comorbid conditions, etc.), then perhaps these high-risk groups have a higher risk for CC398 SAB than do other people. 

2.     Lines 187-192. This conclusion, that the MBEC of cloxacillin tends to be higher among strains carrying the phiMR-11 like prophage, is based on a very small number of isolates.  Of course, there could be many alternative reasons for the variability in MBEC of cloxacillin. This is not convincing evidence of a true association.  

3.     While the genetic variability among CC398 BSI strains in this study does support the idea that these strains were not shared among patients who developed BSI, it does not disprove the idea that the CC398 strains were acquired by different patients in the medical setting independently and subsequently went on to cause BSI. In other words, there could be multiple CC398 hospital strains. 

Minor comments:

1.     Line 50. “emerging” is written twice in this line. 

2.     Line 77. I believe that there is a missing paren “)” on this line. 

3.     How were the 1 of 10 ICUs chosen for the microbiological study?

4.     Table 1, footnote, lines 123-124. “the 242 patients…” Are these the 242 patients with S. aureus bacteremia?

5.     Table 2.  What were the criteria for “Immunosuppression”? Can these criteria be added to the footnote?

6.     Table 2. “Severity score IGS II (adults)2”. Is this 2 intended to be a superscript?

7.     Line 143. “others” would be better “other”

8.     Line 143 and elsewhere in the manuscript. “mupA”, “qacAB”, and “qacC” – if these are gene names, should they be italicized?

9.     Line 148. “tst” – is this the tst1 gene? If so, should it be in italics?

10.  Table 3. “clowacillin” should be “cloxacillin”; “tetracyclin” should be “tetracycline”

11.  Line 162. “eradication” is misspelled.

12.  Lines 204-205. “a large-scale situation” is awkward.  Can the authors consider rephrasing?

13.  Line 220-222. chp and scn and seb – should these gene names be italicized?

14.  Line 227. Gene names – should they be in italics?

Thank you again for the opportunity to review this manuscript. 

Author Response

Reviewer 1

Our respons

Thank you for the opportunity to review the manuscript “Strong biofilm formation and low cloxacillin susceptibility in biofilm-forming CC398 Staphylococcus aureus responsible for bacteremia in French intensive care units, 2021,” by Nathalie van der Mee-Marquet, et al.

The authors performed a study of nosocomial  S. aureus bacteremia (SAB) in 267 French ICUs for 3 months in 2021. There were 1,931 nosocomial bacteria cases recorded. They found that there were 242 cases caused by S. aureus, of which 7.9% were caused by MRSA strains. They chose 10% of the ICUs to perform a more intensive study of SAB and the bacterial isolates causing it. Among the 53 isolates examined, 37.7% were CC398, 17% were CC8, and 13.2% were CC45, with these 3 groups accounting for about 2/3 of all isolates. The percentage of S. aureus belonging to CC398 was remarkably high. Mortality within 7 days of diagnosis among those with CC398 SAB was surprisingly high, at 47.4%. Only 9.4% of the S. aureus isolates were MRSA strains. When glucose was added to growth medium, 62.3% of the isolates formed biofilm, an increase from only 9.4% in the absence of glucose supplementation.  CC398 strains were more likely to form biofilm with cloxacillin broth supplementation compared with glucose supplementation. They also identified a specific prophage more common in strains that were biofilm-formers, the phi-MR11-like prophage.

The authors present data to suggest that biofilm is protective in CC398 MSSA isolates against killing by cloxacillin and that biofilm may be induced by glucose exposure and cloxacillin exposure.

The genetic variability among CC398 strains suggests, the authors report, that these strains may have been introduced from community reservoirs before they went on to cause bacteremia. 

All of the conclusions would be better supported by a larger sample size of S. aureus isolates from SAB.

The sample is unfortunately very modest, and thus the conclusions can only be considered quite tentative. A larger sample size would enable evaluation for statistical significance of the findings comparing CC398 strains to others and also comparing strains with differing phage content. 

We agree with this comment and has indicated this fact in the result section (line 132).

For reviewer’s information, we have analyzed all the strains available from different ICUs located in most regions of France, but in the text of the document, we clearly limited findings to the described situation.

I have several questions and comments for the authors. 

1.  Did the authors test to determine if CC398 was an independent risk factor for death at 7 days?  

The mortality rate among CC398 cases was remarkably elevated, but if these cases occurred disproportionately in patients who had other known risk factors for death (older age, comorbid conditions, etc.), then perhaps these high-risk groups have a higher risk for CC398 SAB than do other people. 

We agree with your remark.

It seems likely that the patients who present a high number of risk factor for death, have a higher risk for CC398 SAB than do other people. Unfortunately, our current data are not able to demonstrate it.  This point has been added into the discussion part (lines 227-228).

2.     Lines 187-192. This conclusion, that the MBEC of cloxacillin tends to be higher among strains carrying the phiMR-11 like prophage, is based on a very small number of isolates.  Of course, there could be many alternative reasons for the variability in MBEC of cloxacillin. This is not convincing evidence of a true association.  

We agree with this comment related to the number of strains. We juste emitted this observation which is in accordance with our results. We currently carry out further study to investigate this hypothesis.

3.     While the genetic variability among CC398 BSI strains in this study does support the idea that these strains were not shared among patients who developed BSI, it does not disprove the idea that the CC398 strains were acquired by different patients in the medical setting independently and subsequently went on to cause BSI. In other words, there could be multiple CC398 hospital strains. 

Based on epidemiological deifinition, the isolates identified in our patients all correspond to nosocomial BSIs acquired in the ICUs. The WGS data allow us to exclude intra-unit patient-to-patient contamination or contamination originating from a unique source.

The design of our study did not allow us to determine where and when the CC398 S. aureus were acquired by the patients. We clarified this point into the discussion section (lines 276-279).

 Minor comments:

1.     Line 50. “emerging” is written twice in this line. 

Modified in the revised version.

2.     Line 77. I believe that there is a missing paren “)” on this line. 

checked.

3.     How were the 1 of 10 ICUs chosen for the microbiological study?

The ICUs were not choosen. All the ICUs were proposed to participate the microbiological study. Finally, 10% of the ICUs participated to this part. Considering HCF categories and location, this set of 30 ICUs was representative of the whole group of ICUs.

4.     Table 1, footnote, lines 123-124. “the 242 patients…” Are these the 242 patients with S. aureus bacteremia?

Yes, these results concerned the 242 patients with S. aureus bacteremia.

5.     Table 2.  What were the criteria for “Immunosuppression”? Can these criteria be added to the footnote?

The presence of immunosuppression is defined in our protocol according to several criteria:

1- either in the presence of true aplasia with <500 circulating polynuclear
2- either the patient is on immunosuppressive treatment (chemotherapy,
radiotherapy, immunosuppressants, long-term and high-dose corticosteroid therapy
dose, eg. for >30 days, recent corticosteroid therapy at a dose >5mg/kg of
Prednisolone for >5 days)
3- either the patient is HIV positive with CD4 < 500/mm 3
or 4- either the patient has leukemia or lymphoma with PNN<500/mm3.

This has been added to table 2 as a footnote in the revised document.

6.     Table 2. “Severity score IGS II (adults)2”. Is this 2 intended to be a superscript?

Modified in the revised version.

7.     Line 143. “others” would be better “other”

Modified in the revised version.

8.     Line 143 and elsewhere in the manuscript. “mupA”, “qacAB”, and “qacC” – if these are gene names, should they be italicized?

Modified in the revised version.

9.     Line 148. “tst” – is this the tst1 gene? If so, should it be in italics?

Modified in the revised version.

10.  Table 3. “clowacillin” should be “cloxacillin”; “tetracyclin” should be “tetracycline”

Modified in the revised version.

11.  Line 162. “eradication” is misspelled.

Modified in the revised version.

12.  Lines 204-205. “a large-scale situation” is awkward.  Can the authors consider rephrasing?

Modified in the revised version.

13.  Line 220-222. chp and scn and seb – should these gene names be italicized?

Modified in the revised version.

14.  Line 227. Gene names – should they be in italics?

Modified in the revised version.

Reviewer 2 Report

In this manuscript, authors described prevalence of S.aurues nosocomial bacteremia in 267 ICUs, revealed the enhanced biofilm formation ability and low cloxacillin susceptibility of CC398 isolates detected in 30 ICUs (11.2%, 30/267). Although, authors did WGS for 20 CC398 strains identified in this study and compared with previously studied collection of 23 CC398 strains, poor presentation in results and discussion, with incomplete procedures in materials and methods section.

 In abstract, line 23, lukF/S should be read as PVL. In line 32, 128mg per L should be read as 128mg/L and it should be corrected throughout the manuscript. The last sentence of “Our data should encourage------------ hospital settings.” meaning is strange, and it should be rephrased.

Material and methods section, procedure of antimicrobial susceptibility testing is not written properly. How many antimicrobials are used? Which method? Broth microdilution or disc diffusion? How did author judge MRSA phenotypically? MIC of cefoxitin?

There are many virulence factors genes in S.aureus such as haemolysins (hla, hlb, hld, hlg), leucocidins (lukDE, lukM, lukS/F-PV), enterotoxins (sea-see, seg-seu, selw, selx, sey, selz, sel26, sel27), tst-1, exfoliative toxins (eta, etb, etd), adhesins (clfA, clfB, fib, fnbA, fnbB, icaA, icaD, ebpS, bbp, eno, cna, etc), modulator of host defense (sak, chp, scn). Why did author choose tst-1 and lukS/F for virulence genes detection by PCR for all the (53) isolates.?

Line 86-87, what did author mean luk genes?  lukM, lukDE, lukS/F-PV?

Line 94, MIC/4 should be read as MIC 4mg/L and correct thought the text.

All the genes names and S.aureus should be italicized and correct in the manuscript.

tst should be read as tst-1 and correct thought the manuscript.

Only 11.2%, 30 out of 267 ICUs were participated in microbiological analysis and this data represents the National Level? Authors should mention it as limitation of this study in the text.

Table 3, according to lineage (MLST) is no meaning, it should be read as ST/CC.

Under subheading of Antimicrobials susceptibility, MRSA is no meaning, it must be antimicrobials showing resistance such as OXA (oxacillin), FOX (cefoxitin). It should be described as Antimicrobials resistance profile listed resistance antimicrobials and drug resistance genes such as mecA, mupA, so on. This reviewer wonders how did they judge Fosfomycin (FOF) resistance? No description in materials and methods section.

Virulence genes, tst and lukF/S should be read as tst-1 and PVL respectively.

clowacillin MIC/4 should be read as cloxacillin MIC (4mg/L).

What is the meaning of MBEC cloxacillin 128??

Figure 1. All the strains should be marked with symbol to discriminate between the current and previous ones or marked with closed circle to isolates of this study. Figure legend is not complete and not understandable by readers.

What is the main purpose to compare the current CC398 isolates and previously isolated strains?

Previously studied 23 CC398 strains, including 6 livestock-associated strains and 17 emerging strains, what did authors mean “emerging strains”? It should be described in the text.

Antibiotype should be read as antimicrobial resistance pattern.

Abbreviation of antimicrobials should be 3 letters code recommended by AAC.

Sa3 and MR-11 should be read as ΦSa3 and ΦMR11respectively.

Author Response

Reviewer 2

Our respons

In this manuscript, authors described prevalence of S.aurues nosocomial bacteremia in 267 ICUs, revealed the enhanced biofilm formation ability and low cloxacillin susceptibility of CC398 isolates detected in 30 ICUs (11.2%, 30/267).

Although, authors did WGS for 20 CC398 strains identified in this study and compared with previously studied collection of 23 CC398 strains, poor presentation in results and discussion, with incomplete procedures in materials and methods section.

In abstract, line 23, lukF/S should be read as PVL.

Authors disagree with this comment. The text describes the presence/absence of gene.

In line 32, 128mg per L should be read as 128mg/L and it should be corrected throughout the manuscript.

Modified in the revised version.

The last sentence of “Our data should encourage------------ hospital settings.” meaning is strange, and it should be rephrased.

The text has been modified (lines 33-34).

Material and methods section, procedure of antimicrobial susceptibility testing is not written properly.

How many antimicrobials are used?

Which method? Broth microdilution or disc diffusion?

How did author judge MRSA phenotypically? MIC of cefoxitin?

The Material and methods section has been clarified (lines 84-87).

There are many virulence factors genes in S.aureus such as haemolysins (hla, hlb, hld, hlg), leucocidins (lukDE, lukM, lukS/F-PV), enterotoxins (sea-see, seg-seu, selw, selx, sey, selz, sel26, sel27), tst-1, exfoliative toxins (eta, etb, etd), adhesins (clfA, clfB, fib, fnbA, fnbB, icaA, icaD, ebpS, bbp, eno, cna, etc), modulator of host defense (sak, chp, scn). Why did author choose tst-1 and lukS/F for virulence genes detection by PCR for all the (53) isolates.?

Authors agree with this comment. However TSST and PVL are the 2 most potent toxins produced by S. aureus. Most of enterotoxins and other factors (adhesins, biofilms, etc..) contribute to virulence but are also very frequently identified in colonizing strains. The vast majority of papers in the fiels report primarily on the presence of these 2 virulence factors.

Line 86-87, what did author mean luk genes?  lukM, lukDE, lukS/F-PV?

Authors agree with this remark and modified accordingly in the document as we reported on the presence of lukS/F (line 88).

Line 94, MIC/4 should be read as MIC 4mg/L and correct thought the text.

MIC/4 indicates a concentration of cloxacillin equal to the strain’s MIC value for cloxacillin, divided by 4. This point has been clarified (lines 160-161, table 3).

All the genes names and S.aureus should be italicized and correct in the manuscript.

tst should be read as tst-1 and correct thought the manuscript.

Modified in the revised version.

Only 11.2%, 30 out of 267 ICUs were participated in microbiological analysis and this data represents the National Level? Authors should mention it as limitation of this study in the text.

We agree it is a limitation of our study and clarified this point into the text (line 132). We continue our research to implement our data.

Table 3, according to lineage (MLST) is no meaning, it should be read as ST/CC.

Under subheading of Antimicrobials susceptibility, MRSA is no meaning, it must be antimicrobials showing resistance such as OXA (oxacillin), FOX (cefoxitin).

It should be described as Antimicrobials resistance profile listed resistance antimicrobials and drug resistance genes such as mecA, mupA, so on.

This reviewer wonders how did they judge Fosfomycin (FOF) resistance? No description in materials and methods section.

Modified in the revised version.

Modified in the revised version.

Regarding fosfomycin, we indicated the results obtained using the AST-P631 Vitek card.

Virulence genes, tst and lukF/S should be read as tst-1 and PVL respectively.

clowacillin MIC/4 should be read as cloxacillin MIC (4mg/L).

Regarding genes we kept lukS/F for the genes encoding PVL, and used tst-1 in the whole revised document.

Regarding the exposure of the strains to cloxacillin, the concentration used varied according to each studied isolate. For a particular isolate, the concentration corresponds to the MIC of this isolate divided by 4.

What is the meaning of MBEC cloxacillin 128??

MBEC stands for minimal biofilm eradication concentration. mg/L has been added in the table 3.

Figure 1. All the strains should be marked with symbol to discriminate between the current and previous ones or marked with closed circle to isolates of this study.

Modified in the revised version.

Figure legend is not complete and not understandable by readers.

What is the main purpose to compare the current CC398 isolates and previously isolated strains?

The comparison of the CC398 recovered in 2021 from ICU patients with the CC398 previously studied was carried out to show the genetic diversity of the novel strains and their distribution among the two major populations within the species (i.e. the livestock-associated ppulation and the human-adapted or emerging clone).

We are working on evolution of CC398 in the clinical settings and this analysis depicted the divergence/relatedness of CC398 on this period of time.

Previously studied 23 CC398 strains, including 6 livestock-associated strains and 17 emerging strains, what did authors mean “emerging strains”? It should be described in the text.

The emerging strains are the strains responsible for BSI in human in an animal-free environment (lines 44-51).

Antibiotype should be read as antimicrobial resistance pattern.

Abbreviation of antimicrobials should be 3 letters code recommended by AAC.

Modified in the revised version.

Modified in the revised version.

Sa3 and MR-11 should be read as ΦSa3 and ΦMR11respectively.

Modified in the revised version.

Round 2

Reviewer 2 Report

Authors modified the manuscript according to reviewers' comments and improvement was noted.